

# Metagenomic insight into taxonomic composition, environmental filtering and functional redundancy for shaping worldwide modern non-lithifying microbial mats

Mariette Viladomat Jasso[1], Manuel García-Ulloa[2], Icoquih Zapata-Peñasco[3], Luis E. Eguiarte[1] and Valeria Souza[1,4]

[1] Departamento de Ecología Evolutiva, Instituto de Ecología, Universidad Nacional Autónoma de México, Ciudad de México, Mexico
[2] ISGlobal, Hospital Clinic, Universitat de Barcelona, Barcelona, Spain
[3] Dirección de Investigación en Transformación de Hidrocarburos, Instituto Mexicano del Petróleo, Ciudad de México, Mexico
[4] Centro de Estudios del Cuaternario de Fuego-Patagonia y Antártica (CEQUA), Punta Arenas, Chile

Corresponding author
Valeria Souza, souza@unam.mx

## ABSTRACT

Modern microbial mats are relictual communities mostly found in extreme environments worldwide. Despite their significance as representatives of the ancestral Earth and their important roles in biogeochemical cycling, research on microbial mats has largely been localized, focusing on site-specific descriptions and environmental change experiments. Here, we present a global comparative analysis of non-lithifying microbial mats, integrating environmental measurements with metagenomic data from 62 samples across eight sites, including two new samples from the recently discovered Archaean Domes from Cuatro Ciénegas, Mexico. Our results revealed a notable influence of environmental filtering on both taxonomic and functional compositions of microbial mats. Functional redundancy appears to confer resilience to mats, with essential metabolic pathways conserved across diverse and highly contrasting habitats. We identified six highly correlated clusters of taxa performing similar ecological functions, suggesting niche partitioning and functional specialization as key mechanisms shaping community structure. Our findings provide insights into the ecological principles governing microbial mats, and lay the foundation for future research elucidating the intricate interplay between environmental factors and microbial community dynamics.

## INTRODUCTION

Microbial mats are multilayered communities considered to be the most primitive communities on Earth, containing in many cases all the known biogeochemical cycles

within only a few millimeters of sediment (*Breitbart et al., 2009*; *Peimbert et al., 2012*). Fossilized material derived from microbial mats can date back as far as 3.7 billion years (*Nutman et al., 2016*).

The typical multilayered structure of microbial mats ('vertical stratification') originates from physicochemical gradients, such as light, oxygen and sulfur, that are generated and maintained by the metabolic activity of their members (*Bolhuis, Cretoiu & Stal, 2014*). These gradients create microenvironments tailored to specific metabolic needs and tolerances of various functional guilds (*Canfield & Des Marais, 1993*; *Van Gemerden, 1993*). From the bottom to the uppermost layer, microbial mats often start with the strictly anaerobic methanogens, followed by the diverse sulfur bacteria and archaea, the early phototrophic guilds of purple and green sulfur and non-sulfur bacteria (*Brocks et al., 2005*; *Oliver et al., 2021*), and finally, cyanobacteria and eukaryotes (*Bolhuis & Stal, 2011*; *Bolhuis, Cretoiu & Stal, 2014*).

Although microbial mats are thought to had been common in the distant past (*Knoll, Bergmann & Strauss, 2016*; *Hamilton, Bryant & Macalady, 2016*; *Lenton & Daines, 2017*), modern microbial mats are found isolated from each other, in locations where environmental conditions are often extreme, and proliferation of algae and most grazing eukaryotic organisms is limited (*Gehling, 1999*; *Kershaw, 2017*). Modern microbial mats can be classified as lithifying (including stromatolite -laminated-, thrombolite -clotted-, dendrolite -dendritic-, leolite -aphanitic- communities) and non-lithifying, depending on the extent of carbon and mineral precipitation they undergo and its resulting structural form (*Dupraz et al., 2009*; *Babilonia et al., 2018*; *Jung & Bowles, 2021*). Examples of sites where lithifying and non-lithifying types of mats thrive include Shark Bay in Western Australia (*Ruvindy et al., 2016*) and the extremely oligotrophic ponds from the Cuatro Ciénegas Basin (*Souza et al., 2008*, *2012*). Nevertheless, there are many sites with non-lithifying mats, including hypersaline sites such as Guerrero Negro (*Javor & Castenholz, 1981*), Rottnest Island (*Mendes Monteiro et al., 2020*) and Great Salt Lake (*Lindsay, Dunham & Boyd, 2020*), or oligotrophic marine sites, like Highborne Cay (*Baumgartner et al., 2009*) and Schiermonnikoog (*Bolhuis, Fillinger & Stal, 2013*). and karstic environments, including Laguna de Bacalar (*Gischler, Gibson & Oschmann, 2008*), Cenote Azul (*Yanez-Montalvo et al., 2021*), Lagunas de Ruidera (*Santos et al., 2010*), Pavillion Lake (*Russell et al., 2014*) and Lake Clifton (*Moore, 1987*; *Warden et al., 2016*). Interestingly, microbial mats have also been found at extreme enviroments, including high temperature sites, as in Little Hot Creek and Yellowstone (*Ward et al., 2006*; *Kraus et al., 2018*; *Wilmeth et al., 2018*), polar regions (*Taton et al., 2003*; *Varin et al., 2012*) and uranium and gold mines (*Drewniak et al., 2016*).

However, research on non-lithifying microbial mats has been performed predominantly at a local scale, often delving into site-specific descriptions, time series analyses, and environmental change experiments (*Green et al., 2008*; *Cardoso et al., 2017*; *Babilonia et al., 2018*; *De Anda, Zapata-Peñasco & Souza, 2018*; *Gutiérrez-Preciado et al., 2018*; *Medina-Chávez et al., 2023*), or comparisons between mats within similar environmental conditions (*Vincent, 2000*; *Andersen et al., 2011*; *Varin et al., 2010*, *2012*). These localized studies have yielded valuable insights into microbial mat ecology, such as the dominant

role of Cyanobacteria on their community ecology (*Ley et al., 2006*; *Cardoso et al., 2017*), their impressive capabilities for nutrient scavenging and recycling (*Varin et al., 2012*), their resilience in the face of perturbations due to their functional redundancy (*Green et al., 2008*; *Babilonia et al., 2018*; *De Anda, Zapata-Peñasco & Souza, 2018*) which extends across lithifying and non-lithifying mat types (*Khodadad & Foster, 2012*), and their high metabolic interdependence (*Paerl, Pinckney & Steppe, 2000*; *Villanueva, Del Campo & Guerrero, 2010*). Moreover, in accordance with many studies that recognize geographic location, temperature, and pH as important drivers of both taxonomic and functional microbial community structure (*Yannarell & Triplett, 2005*; *Van der Gucht et al., 2007*; *Frindte et al., 2019*; *Jiao & Lu, 2020*), a few studies in microbial mats have also found these environmental factors to play an important role in community composition (*Peimbert et al., 2012*; *De Anda, Zapata-Peñasco & Souza, 2018*; *Uribe-Lorío et al., 2019*).

Despite these insights, comprehensive global comparisons of microbial mats incorporating environmental measurements remain scarce. With only a few reviews and a single comparative study lacking environmental data (*Bolhuis, Cretoiu & Stal, 2014*; *Prieto-Barajas, Valencia-Cantero & Santoyo, 2018*; *Santoyo, 2021*), there is a notable gap in our understanding of microbial mat dynamics at a broader scale.

Drawing on the concept of environmental filtering, which underscores how the environment selectively shapes community assembly and function by eliminating species unable to withstand specific conditions at a given location (*Kraft et al., 2015*), we hypothesized that environmental differences among sites will significantly influence both the taxonomic and functional composition of microbial mats. Therefore, each site will have a taxonomically and functionally unique microbial mat, with greater divergence correlated with both increased environmental differences and geographic distance. To test our hypothesis, we performed diversity and statistical analyses on 62 publicly available non-lithifying microbial mat Illumina MiSeq shotgun metagenomes with geographic coordinates, and environmental measurements of temperature and pH from eight sites across the globe, including a newly found hypersaline mat from Cuatro Ciénegas in Mexico named Archaean Domes.

## MATERIALS AND METHODS

### Sampling, sequencing and assembly of Archaean Domes metagenomes

Samples were taken under SEMARNAT scientific permit SGPA/DGVS/03121/15 from a recently discovered pond named "Archaean Domes" in Rancho Pozas Azules, located at the southeast of the Sierra de San Marcos within Cuatro Ciénegas Basin, (CCB, 26° 49′ 41.7″ N 102° 01′ 28.7″ W), in the state of Coahuila, in Northeast Mexico.

Sampling took place on April 10, 2016, during dry season and September 3, 2016, during wet season. Water physicochemical conditions were measured 5 cm deep during the second sampling period, using a Hydrolab MS5 Water multiparameter probe (OTT Hydromet GmbH, Germany) (Table S1). During the first sampling period the ponds were dry, so no physicochemical measurements could be taken. From each sampling period,

$5 \text{ cm}^3$ of microbial mat samples were obtained using a sterilized scalpel and transferred to sterile Falcon tubes (50 mL) and immediately placed in liquid nitrogen before being stored at −80 °C to preserve the structural integrity of the mat until DNA processing.

Images of Archaean Domes mats during these samplings can be found on *Medina-Chávez et al. (2023)* and further description of the site on *Espinosa-Asuar et al. (2022)* and *Madrigal-Trejo et al. (2023)*.

The DNA extraction from the Archaean Domes samples was performed using a column-based modification of the protocol described by *Purdy et al. (1996)*. Shotgun whole-genome sequencing, employing paired-end $2 \times 300$ sequencing on an Illumina MiSeq platform, was conducted at CINVESTAV-LANGEBIO, Irapuato, Mexico. Raw metagenomic reads from these samples, as well as from an additional sample collected in February 2017 during an extremely dry season (not included in this study), are publicly available at NCBI through the BioProject PRJNA612690.

Bioinformatic tools used for this study were executed with their default parameter settings, unless explicitly specified. Raw data from Archaean Domes was quality-checked using FastQC v.0.11.9 (*Andrews, 2010*). Indexed adapters and barcodes were removed, and low quality sequences were discarded with Trimmomatic v.0.36 using a sliding window of 4 pb and an average quality-per-base of 25 (*Bolger, Lohse & Usadel, 2014*). Trimmed reads were uploaded to MG-RAST platform under the project ID mgp95137.

## Metagenome selection

In addition to our microbial mat samplings, we also selected 60 publicly available shotgun metagenomes of non-lithifying (as stated by their metadata and publications, when available) microbial mats from the MG-RAST repository. To achieve a robust metagenomic comparison in terms of completeness, sequencing depth and hit length, we verified that all analyzed metagenomes were sequenced with the Illumina MiSeq sequencing platform and submitted as raw reads. Moreover, sufficiency of sampling effort was assessed with rarefaction curves of species richness against number of reads (Fig. S1). To allow for interpretability, comparability and reproducibility, metagenomes that did not meet these criteria were excluded from the analysis.

In total, 62 shotgun metagenomes from eight sites from across the globe were used: Archaean Domes CCB (Mexico; n of metagenomes = 2; MG-RAST project ID: mgp95137); Shark Bay (Australia; $n = 15$; mgp82056) (*Ruvindy et al., 2016*); Little Hot Creek (USA; $n = 4$; mgp7509 and mgp9770) (Spear, 2014 from MG-RAST); Schiermonnikoog (Netherlands, $n = 17$; mgp83560) (Bolhuis, 2015 from MG-RAST); Death Valley (USA, $n = 6$; mgp20082) (*Stamps et al., 2018*); Mono Lake (USA, $n = 3$; mgp19115) (Spear, 2016 from MG-RAST); Rottnest Island (Australia, $n = 10$; mgp84373) (*Mendes Monteiro et al., 2020*); and Kowary and Zloty Stok mines (Poland, $n = 3$ and 2, respectively; mgp16411) (*Drewniak et al., 2016*) (Table 1).

MG-RAST accession numbers and coordinates of all metagenomes used in this study can be found in Table 1, and the temperatures and pH measurements of each site in Table S1. For metagenomes without scientific publication, MG-RAST metadata was used, when available.

**Table 1 Sampling of metagenomes from MG-RAST.**

| Site | n | General description | Coordinates (latitude, longitude) | MG-RAST project ID | Sample names | Mean hits (taxonomy) | Mean hits (function) | Publication |
|------|---|---------------------|-----------------------------------|--------------------|--------------|----------------------|----------------------|-------------|
| Archaean Domes | 2 | Oligotrophic | 26.82825, −102.024639 | mgp90438 | ADM1_R1_q25, ADM2_R1_q25 | 1,151,456 ± 869,896 | 416,838 ± 320,161 | This study |
| Little Hot Creek | 4 | Hot spring | 37.694518, −118.8165 | mgp7509; mgp9770 | GB14.Black.Soil.Upstream.of.Weir, GB14.Cone.Pool.Mat, GB14.Orange.Biofilm.Below.Weir, LHC14.Cone.Pool.Cones | 5,235,831 ± 1,419,251 | 2,430,818 ± 665,768 | J. Spear, 2014, from MG-RAST |
| Schiermonnikoog | 17 | Coastal | 6.122539, 53.472543 | mgp83560 | M1_1, M1_2, M2_2, M3_1, M3_2, M4__2, M4_1, M5_1, M5_2, M6_1, M7_1, M7_2, M8_1, M8_2, M9_2 | 4,277,949 ± 200,831 | 1,639,988 ± 109,669 | H. Bolhuis, 2015, from MG-RAST |
| Shark Bay | 15 | Hypersaline | −26.421, 114.279 | mgp82056 | SD10_S10_L001_R1_001, SD3_S15_L001_R1_001, SD6_S2_L001_R1_001, SD6_S2_L002_R1_001, SD7_S4_L001_R1_001, SSD10_S12_L002_R1_001, SSD10_S12_L003_R1_001, SSD3_S16_L003_R1_001, SSD4_S18_L001_R1_001, SSD4_S18_L002_R1_001, SSD6_S3_L003_R1_001, SSD7_S5_L001_R1_001, SSD8_S7_L003_R1_001, SSD8_S7_L004_R1_001, SSD9_S9_L002_R1_001 | 459,728 ± 155,862 | 184,385 ± 63,377 | *Ruvindy et al. (2016)* |
| Kowary | 3 | Mine wastewater | 50.784766, 15.841175 | mgp16411 | KOW_0, KOW_2, KOW_3 | 21,560,149 ± 6,559,772 | 8,626,099 ± 24,91,289 | *Drewniak et al. (2016)* |
| Zloty Stok | 2 | Mine wastewater | 42.884356, 19.158789 | mgp16411 | Zloty_Stok_ZS_2, Zloty_Stok_ZS_3 | 13,420,669 ± 57,30,526 | 5,225,336 ± 22,75,387 | *Drewniak et al. (2016)* |
| Rottnest | 10 | Hypersaline | −31.999838, 115.51448 | mgp84373 | JM15.R1 , JM23.R1, JM26.R1, JM29.R1, JM30.R1, JM31.R1, JM4.R1, JM5.R1, JM6.R1, JM8.R1 | 14,98,556 ± 203,489 | 540,985 ± 75,876 | *Mendes Monteiro et al. (2020)* |
| Mono Lake | 3 | Alkaline | −119.023778, 37.993513 | mgp19115 | G5088, G5089, G5090 | 4,019,436 ± 547,117 | 1,317,180 ± 166,241 | Spear (2016) from MG-RAST |
| Death Valley | 6 | Hypersaline | 36.229188, −116.768443 | mgp20082 | DEVA_BWPP7_G, DEVA_BWPP7_H, DEVA_BWPP7_I, DEVA_BWPP7_L, DEVA_BWPP7_M, DEVA_BWPP7_N | 15,438,721 ± 4,810,730 | 1,918,323 ± 503,876 | *Stamps et al. (2018)* |

## Metagenomic annotation

Both taxonomic and functional annotations were performed within the MG-RAST platform, using pipeline v.4.03.1a (https://Github.com/MG-RAST/pipeline), which consists of two steps:

1) Metagenomic sequences were preprocessed using SolexaQA v1.2 (*Cox, Peterson & Biggs, 2010*) to trim low-quality regions from FASTQ data, dereplicated with a k-mer

approach (*Gomez-Alvarez, Teal & Schmidt, 2009*) and screened for contaminants with bowtie2 v2.3.4 (*Langmead et al., 2009*).

2) Taxonomic and functional annotations were done using the Lowest Common Ancestor algorithm (*Wood & Salzberg, 2014*) utilizing the RefSeq and KO (Kyoto Encyclopedia of Genes and Genomes Orthology) as reference databases. To define true hits, we used thresholds similar to previous comparative short-read shotgun metagenomics studies but with a longer minimal alignment (*Koo et al., 2016*; *Rabelo-Fernandez et al., 2018*; *Angeli et al., 2019*; *Kibegwa et al., 2020*; *Akinola, Ayangbenro & Babalola, 2021*; *Mohamed et al., 2023*); hits of >25 bases and >65% similarity to a taxonomic group or a function with an E-value of ≤10−5 were included in this study.

## Data processing and analysis

Taxonomic and functional categories with <4 hits and <20% prevalence across samples were removed from the analysis to minimize false positives. In order to make highly variable data comparable, resulting tables were rarefied to the minimum library size and further normalized by the Total Sum Scaling (TSS) method, which has proven highly accurate for shotgun data (*Norouzi-Beirami et al., 2021*).

Statistical analyses were performed within the Microbiome Analyst platform version 1 (https://Github.com/xia-lab/MicrobiomeAnalystR/releases/tag/v1.0.0, *Chong et al., 2020*). All scripts used for statistical analyses and visualization used in this study can be found on the following GitHub and Zenodo repositories: https://Github.com/MarietteViladomat/Worldwide-modern-microbial-mat-METAGENOMICS; https://doi.org/10.5281/zenodo.8305877.

Alpha diversity was calculated using the Shannon index and a Kruskal-Wallis test for multiple groups (version from R base) was performed to test differences in mean values between sites. Beta diversity was calculated from Bray-Curtis dissimilarity matrices, and visualized through Non-Metric Multidimensional Scaling (NMDS) plots. Site differences were also assessed quantitatively through analyses of similarities (ANOSIM). Core community analysis was performed with a minimum sample prevalence of 25% and minimum relative abundance of 0.01%. The Linear Discriminant Analysis Effect Size (LEfSe) algorithm implementation from Microbiome Analyst was used to detect features with significantly differential relative abundance across sites. A *P*-value cutoff of 0.01 after False Discovery Rate adjustment was set as the significance threshold.

In addition, an Euclidean distance matrix of the physicochemical environmental measurements and a Harvesine distance matrix of the geographical coordinates of the sites were also calculated with *vegan* and the R package *geosphere v1.5-10* (*Hijmans, Williams & Vennes, 2019*), respectively.

Correlations between beta diversity and geographic distance, as well as environmental measurements of temperature and pH, were calculated with the implementation of the Mantel test from *Legendre & Legendre (2012)* (9,999 permutations, Spearman correlation method). Correlation plots of the Spearman's rho from the top 100 most abundant taxa and functions were drawn with the R package *corrplot v0.84* (*Wei et al., 2017*).

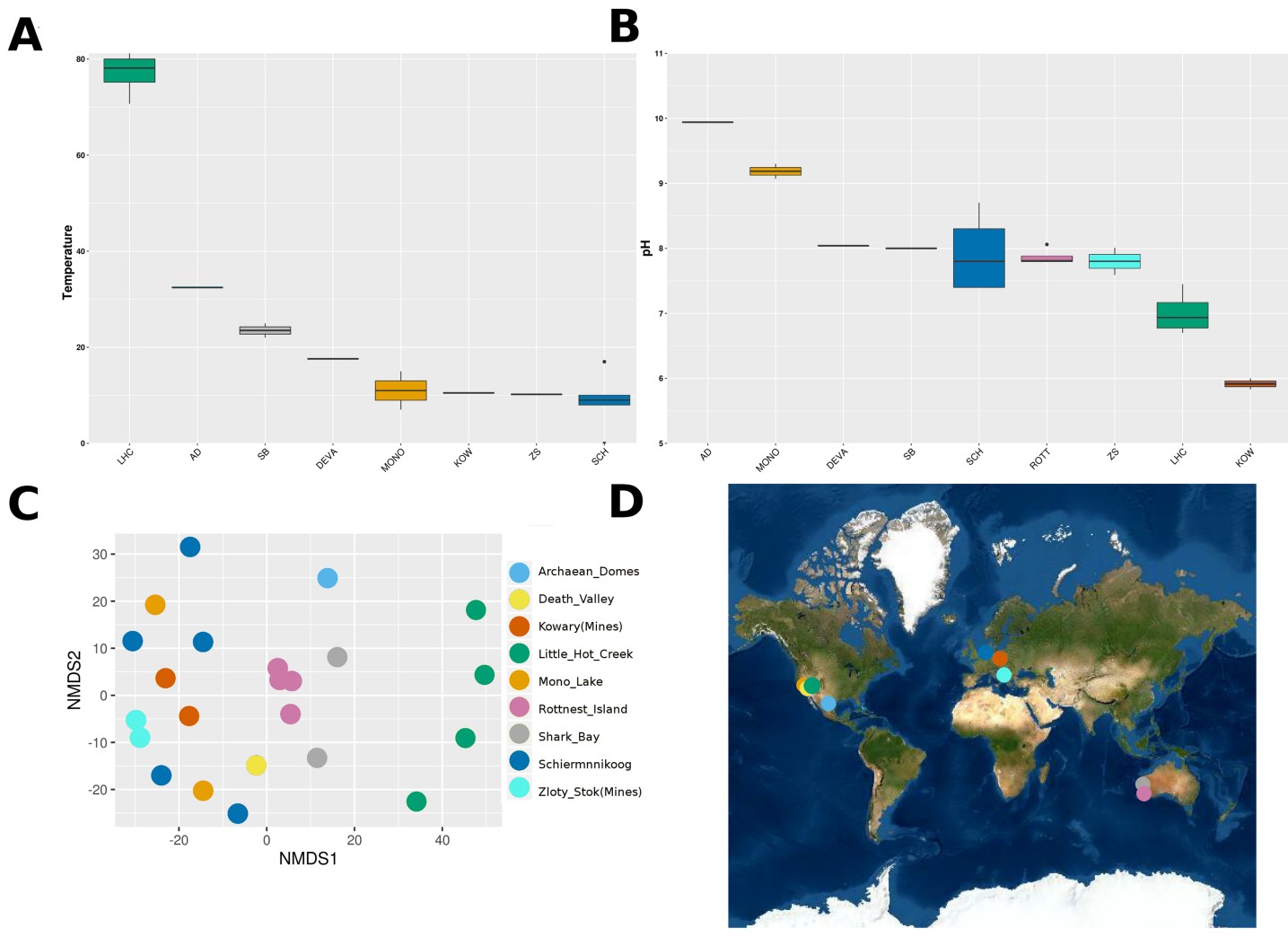

**Figure 1 Environmental conditions and geographical location of microbial mats.** (A) Average temperature (B) average pH. Sites are sorted descending according to their mean value. Individual values of environmental measurements can be found in Table S1. (C) NMDS of environmental variables by the Euclidean distance of temperature and pH as a compound variable (ANOSIM9S R = 0.78, $p < 0.01$; stress score = 0.12). (D) Geographic location of each microbial mat. Sites are colored as follows: Archaean Domes (AD, light blue); Death Valley (DEVA, yellow); Kowary Mines (KOW, red); Little Hot Creek (LHC, green); Mono Lake (MONO, orange); Rottnest Island (ROTT, pink); Shark Bay (SB, gray); Schier-moonnikog (SCH, dark blue); and Zloty Stok mine (ZS, cyan). Rottnest is missing from the temperature plot as there were no measurements available.

# RESULTS

## Description of the sites

Temperature and pH measurements were independent from each other (Spearman's rho = −0.181, $p$-value = 0.458). Measurements of pH went from slightly acidic to alkaline, with Kowary being the most acidic (5.9 ± 0.12) and Archaean Domes the most alkaline (9.94). Physicochemical environmental variables showed consistent sample clustering by geographical site, except for Schiermonnikoog and Mono Lake (Fig. 1). Furthermore, measurements from Little Hot Creek grouped separately from the rest as a result of both their high temperature (77.05 ± 4.628 °C). Contrastingly, Zloty Stok and Kowary grouped

together and were also found at geographic proximity, leading us to consider metagenomes from those sites under the same "Mines" category from now on.

At the genus level, Shark Bay and Schiermonnikoog metagenomes showed a similar pattern of variation, as well as Archaean Domes, Death Valley and Mono Lake (Fig. 2A). However, notable differences occurred among Mines, Little Hot Creek and Rottnest samples: Mines showed a general lower diversity and higher abundance of Candidate *Solibacter*; Little Hot Creek also had lower diversity along a higher abundance of *Salinibacter*, *Roseiflexus*, *Chlorobium*, *Chloroherpeton* and *Anabaena*; Rottnest had a high abundance of *Salinibacter* and *Rhodothermus* (Fig. 2A).

In contrast, KEGG metabolism categories (Fig. 2B) were similar across sites and samples, with the exception of one sample from Mines, which had a spike in the metabolism of terpenoids and polyketides (Fig. 2B).

## Taxonomic and functional diversity of microbial mats

Mean Shannon's diversity at the genus level was significantly different across sites, ranging from 3.75 to 5.75 (Fig. 3A; Kruskal-Wallis statistic = 52.0, $p$ = 5.8E-9). Most diverse microbial mats were those from hypersaline Death Valley and Mono Lake, closely followed by Shark Bay, while Little Hot Creek had the lowest alpha diversity. Contrastingly, alpha diversity of functions (Fig. 3B) showed a narrower range, albeit significantly different across sites, between 4 and 4.18 (Krustal-Wallis statistic = 26.9, $p$ = 3.3E-4), with the exception of one sample from Mines, which had the lowest Shannon value of 3.88. Noteworthy, samples from Mines had both the highest and lowest functional diversity.

NMDS plots of genera (Fig. 4A) showed a mostly consistent grouping of samples into their corresponding sites, except for Death Valley, which formed two separate groups. Also, Mines and Little Hot Creek samples appear separated from the rest of the sites, displaying low alpha diversity (Fig. 4C).

Contrastingly, grouping of functions was less clear (Fig. 4B). Most samples from Mines, Little Hot Creek, Archean Domes and Schiermonnikoog, as well as some from Rottnest and Shark Bay appeared all scattered across. Interestingly, there was a group of highly diverse samples located at the bottom-middle of the plot (Fig. 4D, blue). The level of diversity decreased (from purple to pink) as the samples moved farther away from this group, especially in the upper-left direction (darkest pink).

## Similarities and differences across microbial mats

The cyanobacterium *Cyanothece* was the most abundant and prevalent member of the microbial mat taxonomic core, with a relative abundance of at least 1% within most of the samples, followed by *Microcoleus* and *Planctomyces* (both with 1% relative abundance in 60% of the samples) and *Rhodopirellula* (1% relative abundance in 50% of the samples) (Fig. S2).

The top 40 most significantly abundant genera found with the LEfSe algorithm (Fig. 5) included *Clostridium* in Mono Lake; *Planctomyces*, *Trichodesmium* and *Rhodopirellula* in Schiermonnikoog; multiple sulfur-related *Desulfovibrio*, *Desulfonatronospira*, *Desulfohalobium* and *Desulfomicrobium* alongside *Halanaerobium*, *Spirochaeta* and

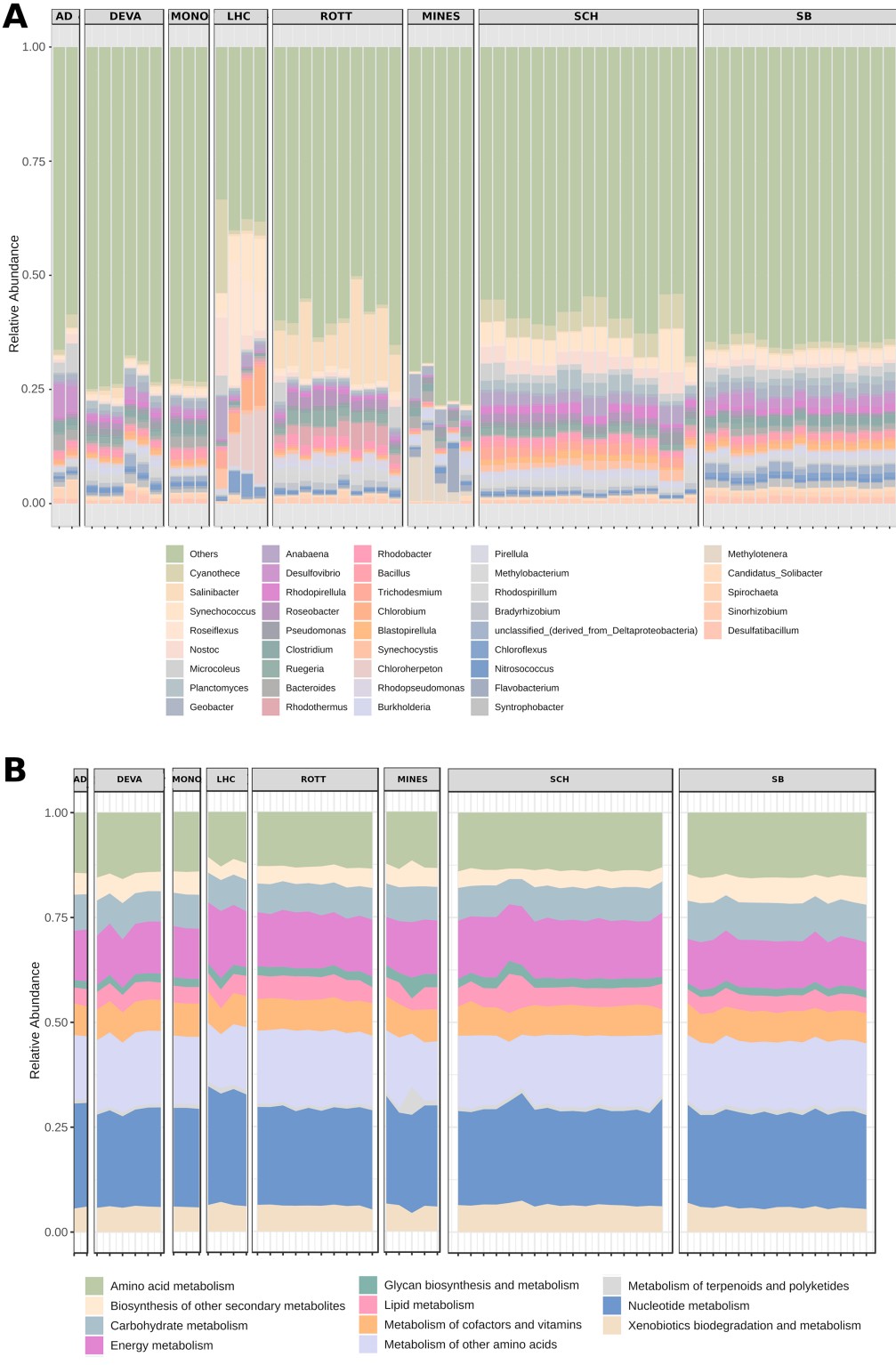

**Figure 2** **Relative abundance of genus and functional categories from microbial mats.** (A) The genus stacked bar plot shows the 40 most abundant genera (B) while the function plot shows all available KEGG metabolism categories. AD, Archaean Domes; DEVA, Death Valley; MONO, Mono Lake; LHC, Little Hot Creek; ROTT, Rottnest Island; MINES, Mines; SCH, Schiermonnikoog; SB, Shark Bay.

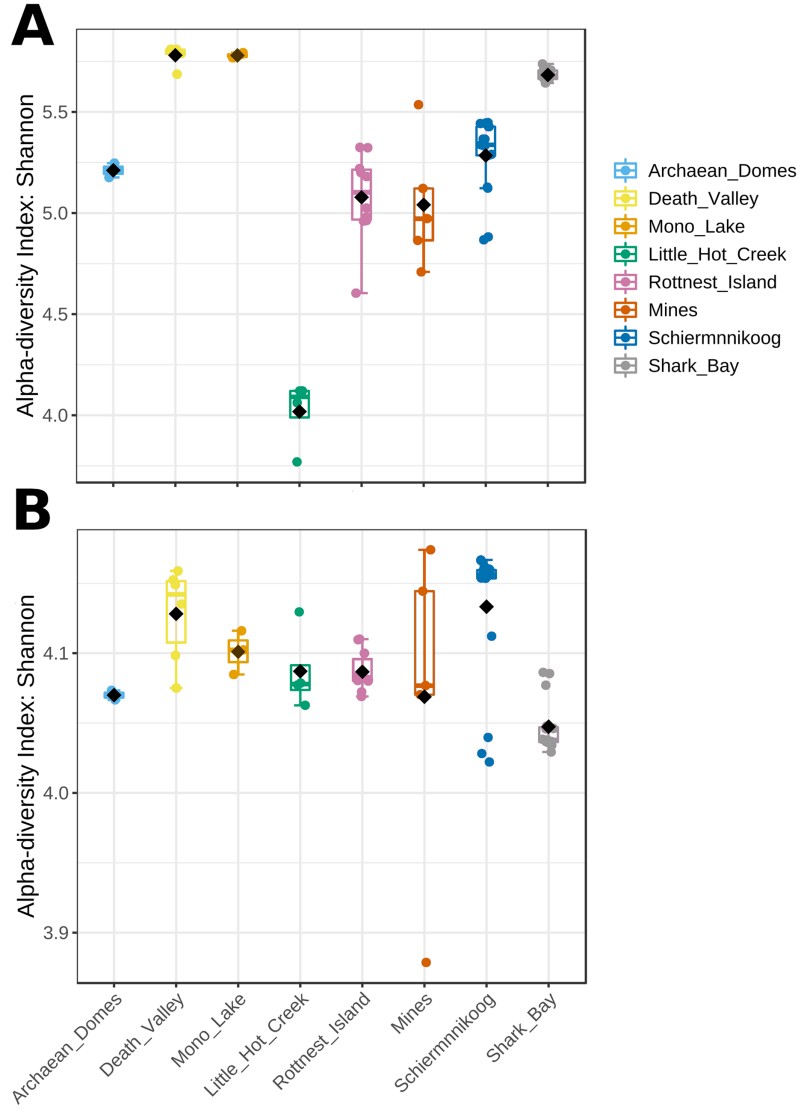

**Figure 3 Alpha diversity from microbial mat communities.** Alpha diversity at (A) genus level and (B) KEGG pathways were calculated with Shannon's Diversity Index. Kruskal-Wallis test gave significant results for both comparisons (statistic = 52.0, $p$ = 5.8E-9 for genus; statistic = 26.9, $p$ = 3.3E-4 for functions).

*Bacteroides* in Archaean Domes; red-pigmented and halophilic bacteria *Rhodothermus, Rhodobacter, Rhodomicrobium* and *Roseobacter, Salinibacter* and *Ruegeria* with *Methylobacterium* in Rottnest Island; methane-related *Methylotenera, Methylobacter, Methylobacillus, Methylococcus,* as well as biodegradation-related generalists including *Geobacter, Pseudomonas, Polaromonas, Albidiferax, Burkholderia, Dechloromonas* and *Xanthomonas* in the Mines cluster; thermophilic bacteria (mostly Cyanobacteria) *Roseiflexus, Synechococcus, Cyanothece, Chlorobium, Nostoc, Anabaena, Chloroflexus, Chlorobaculum* and *Synechocystis* in Little Hot Creek.

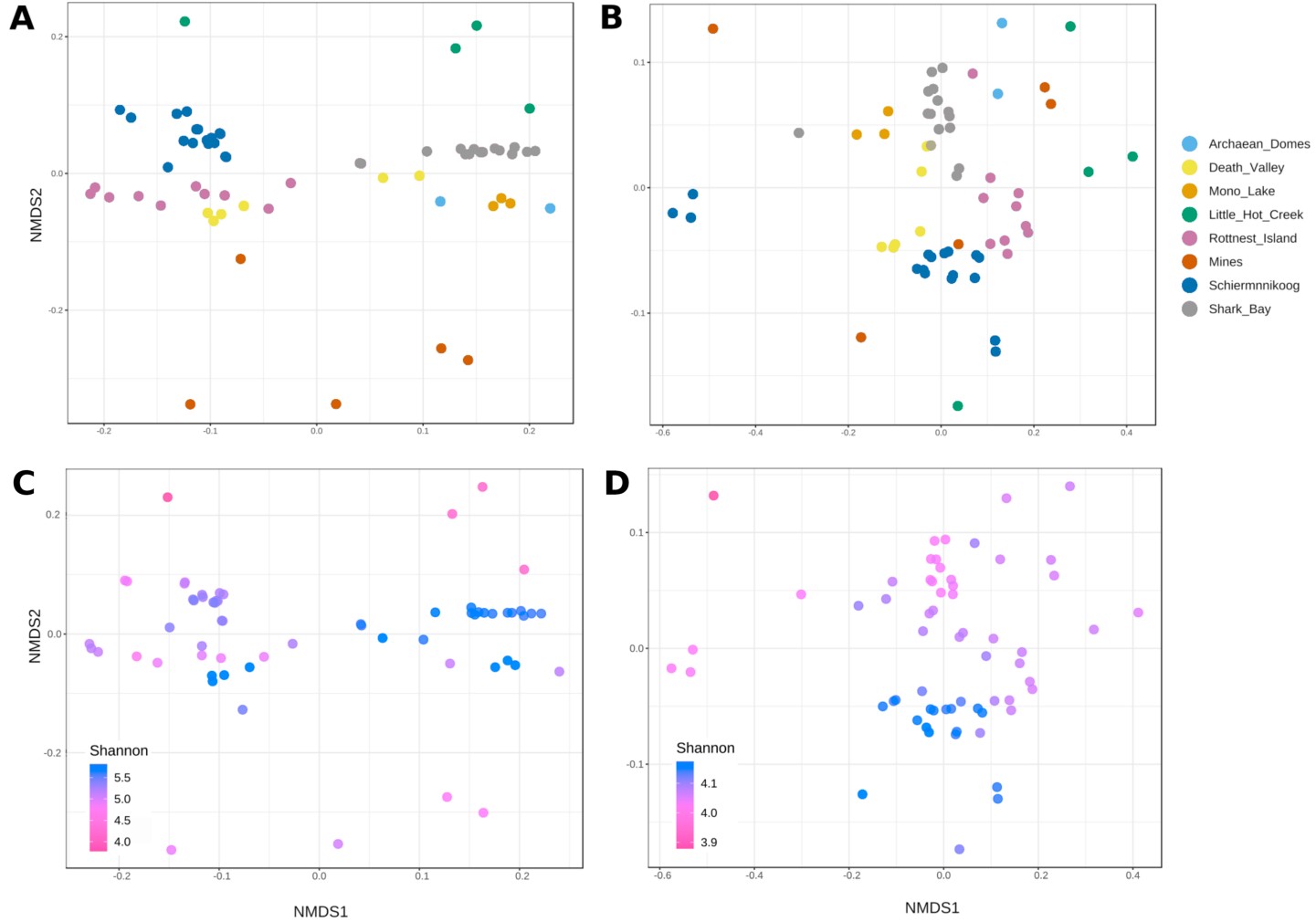

**Figure 4 Non-metric multidimensional scaling of taxonomic and functional beta diversity of microbial mats.** Bray-Curtis dissimilarity matrix was used as a beta diversity metric. (A and C) correspond to genera (ANOSIM's R = 0.91; $p < 0.001$; stress score = 0.1) and (B and D) to functions (ANOSIM's R = 0.67; $p < 0.001$; stress score = 0.08). Bottom panels include Shannon diversity index values as a color gradient to visualize the relationship between alpha and beta diversities.

From the top 100 most abundant genera, a total of six clusters of highly positively correlated bacteria were found across all metagenomes (Fig. 6). Cluster 1 encompasses mostly sulfate-reducing and green sulfur and non-sulfur bacteria, but also well-known spore-forming generalists such as *Bacillus* and *Clostridium*, some members of the Bacteroidaceae family (*Bacteroides*, *Geobacter*) and Spitochaeta. Cluster 2 includes metabolically diverse bacteria with large genomes from the Planctomycetaceae (*Planctomycetes*, *Pirellula*, *Rhodopirelilla*, *Blastopirellula*), Actinobacteria (*Streptomyces* and *Mycobacterium*), myxobacteria (*Anaeromyxobacter*, *Myxococcus*, *Sorangium*) and Acidobacteria (*Solibacter* and *Haliangium*) taxa. Cluster 3 includes diverse and metabolically flexible soil bacteria from the Flavobacteriaceae and Bacteroidota (mostly Cytophagales) taxa. Cluster 4 includes representatives from the purple sulfur bacteria

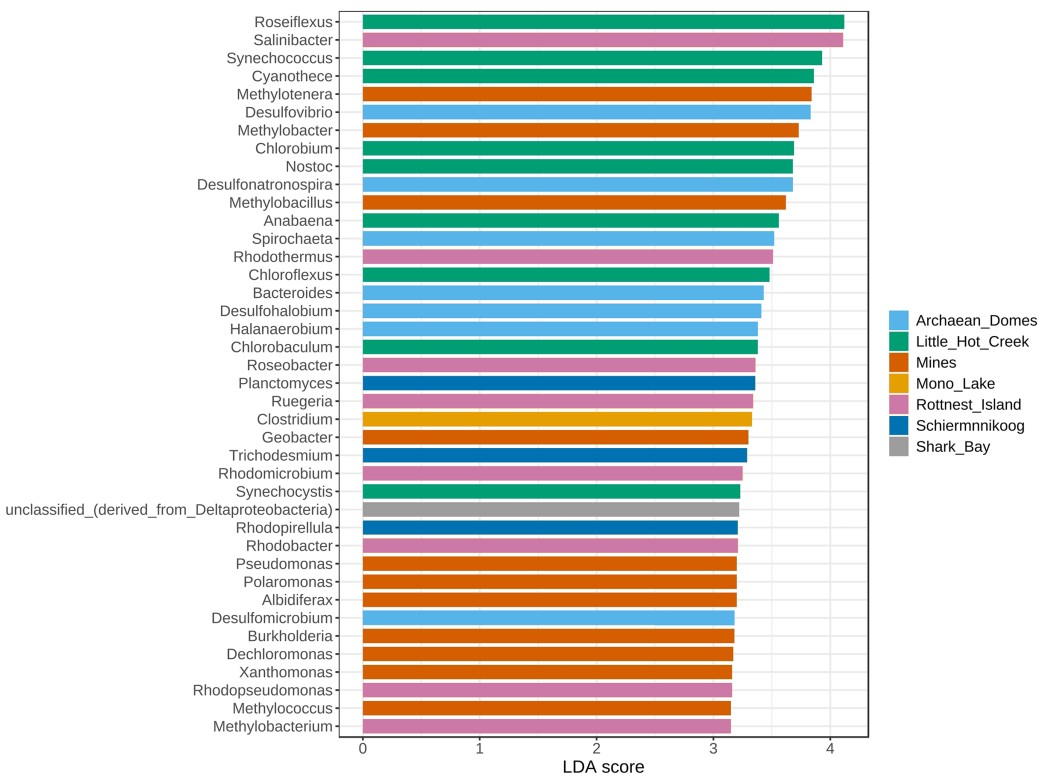

**Figure 5 Linear discriminant analysis effect size (LEfSe) of genera.** Only the top 40 most significant genera are depicted, sorted descendingly by linear discriminant analysis (LDA) score. This particular implementation of the LEfSe algorithm employs Kruskal-Wallis rank sum test to detect features (genera) with significant differential abundance with regards to class labels (sites). *P*-value cutoff of 0.01 after False Discovery Rate adjustment.

group (*Allochromatium*, *Alkalilimnicola* and *Thialkalivibrio*), many metabolically diverse free-living generalists (*Shewanella*, *Vibrio*, *Pseudomonas*, *Burkholderia*, *Xanthomonas*, *Polaromonas*, *Cupravidus*) and members of Methylobacteraceae. Cluster 5 is formed exclusively by Cyanobacteria. Finally, cluster 6 includes bacteria from three main groups: Rhodobacteraceae, Caulobacterales and nitrogen-fixing bacteria.

The correlation plot of the top 100 most abundant functions (Fig. S3) showed two positively correlated large clusters that are also negatively correlated between each other. The top left cluster mainly contains functions related to degradation, particularly of dioxin, polycyclic aromatic hydrocarbons, naphthalene and aromatic compounds, as well as microbial metabolism in diverse environments. Contrastingly, the bottom right cluster is composed of biosynthesis pathways, particularly of steroids, terpenoids and secondary metabolites. A detailed view of each function at lower hierarchical categories can be found in Table S2.

Mantel tests of genera and functions against a compound variable of temperature and pH and a Harvesine distance matrix of geographical coordinates were performed. Of the four possible combinations, only the correlation between functional dissimilarity and geographic distance was significant, *i.e.*, closer mats have a more similar functional structure (Mantel's r = 0.427, *p* = 0.0111; Fig. 7, Table S3).

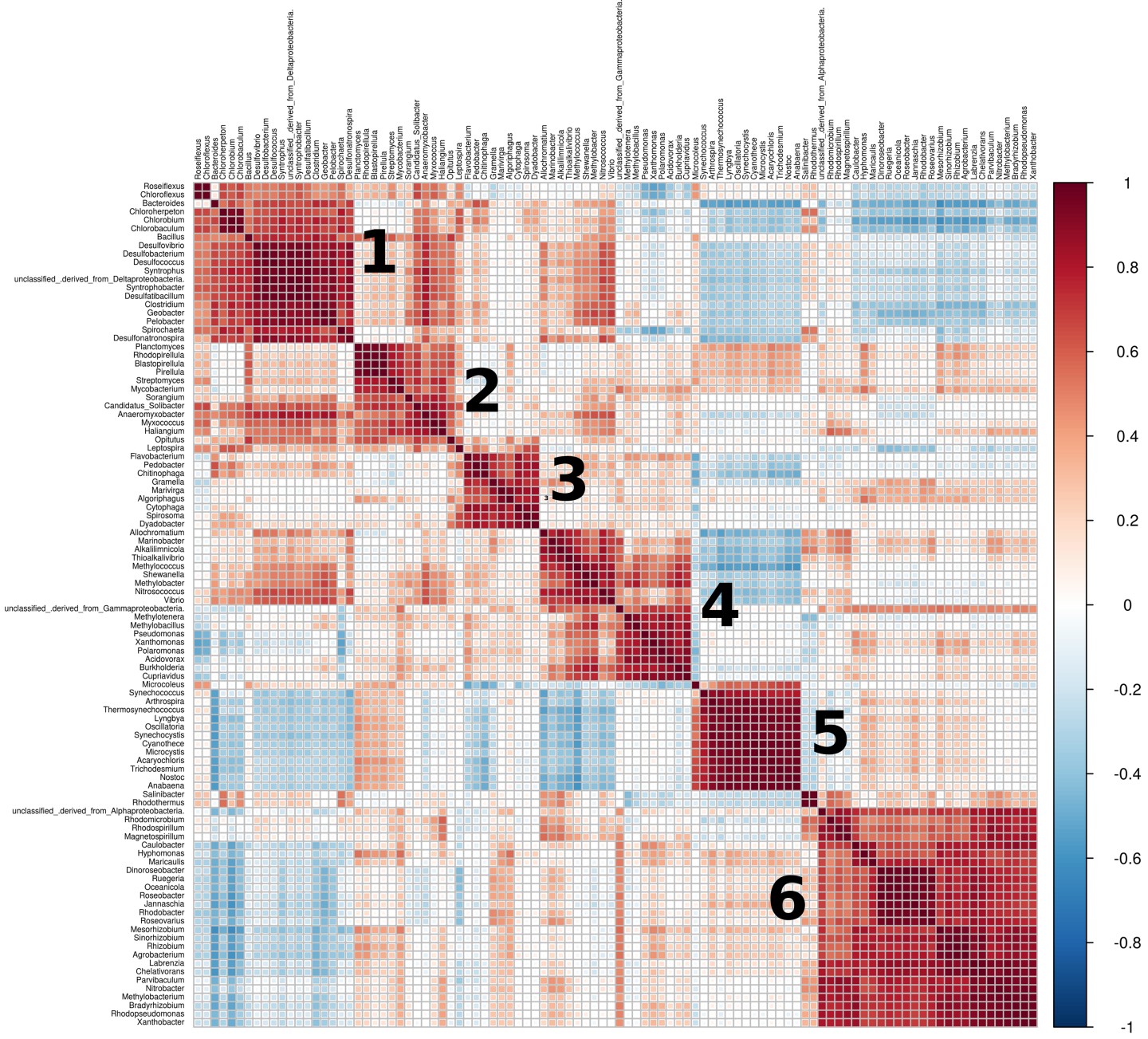

**Figure 6  Correlation plot of relative abundances (taxa).** Spearman's correlation test was performed on the relative abundance of the top 100 most abundant genera. Six clusters of strong positively correlated bacteria were found. Roughly, they can be cataloged as cluster 1, sulfate-reducing, green sulfur and non-sulfur bacteria; cluster 2, Planctomycetaceae, Actinobacteria, myxobacteria and Acidobacteria; cluster 3, Flavobacteriaceae and Bacteroidota; cluster 4, purple sulfur bacteria, Methylobacteraceae and generalists such as Pseudomonas and Vibrio; cluster 5, Cyanobacteria, cluster 6: Rhodobacteraceae, Caulobacterales and nitrogen-fixing bacteria.               

# DISCUSSION

Microbial mats are known to have a high degree of functional redundancy (*Green et al., 2008*; *Babilonia et al., 2018*; *De Anda, Zapata-Peñasco & Souza, 2018*), metabolic interdependence (*Paerl, Pinckney & Steppe, 2000*;

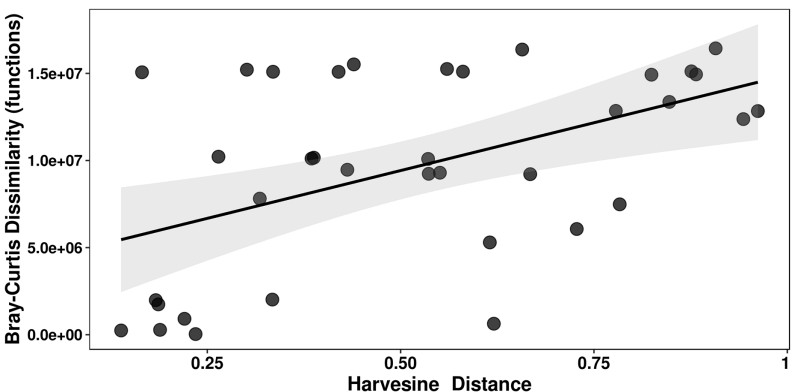

**Figure 7 Mantel test of functions against geographic distance.** A Bray-Curtis dissimilarity matrix and a Harvestine distance matrix were calculated for the relative abundance of functions and the geographic distance (coordinates), respectively. The correlation between the two was significant (Mantel's r = 0.427; $p$ = 0.0111).

*Villanueva, Del Campo & Guerrero, 2010*), and were highly abundant worldwide in the distant past (*Knoll, Bergmann & Strauss, 2016*; *Hamilton, Bryant & Macalady, 2016*; *Lenton & Daines, 2017*). We hypothesized that the environment (pH, temperature and geographic location) strongly shape the structure of modern non-lithifying microbial mats both in terms of taxonomic and functional compositions (see also *Yannarell & Triplett, 2005*; *Van der Gucht et al., 2007*; *Frindte et al., 2019*).

## The influence of the environment

Alpha and beta diversities for both taxonomic and functional categories varied significantly between sites, indicating a strong environmental influence across microbial mats. The most taxonomically diverse mats were also the most saline: Death Valley, Mono Lake and Shark Bay. In contrast, the least diverse mat was the one with higher temperature: Little Hot Creek (Figs. 1, 2, 3A). This aligns with previous findings that salinity and environmental fluctuations are major drivers of microbial diversity in coastal mats (*Bolhuis, Fillinger & Stal, 2013*; *Bolhuis, Cretoiu & Stal, 2014*). Moreover, previous studies have found a negative correlation between diversity and environmental conditions, such as high temperature, low salinity and lack of seasonality (*Everroad et al., 2012*; *Bolhuis, Cretoiu & Stal, 2014*). These set of environmental conditions are found at Little Hot Creek, and would explain the low alpha and beta diversity of its mats.

Contrastingly, even though we still found functional diversity to be significantly different between sites, the observed differences have a narrower range compared to taxonomic diversity (Figs. 3A *vs*. 3B) which is consistent with the known high functional redundancy of microbial mats (*Green et al., 2008*; *Babilonia et al., 2018*; *De Anda, Zapata-Peñasco & Souza, 2018*). These findings are mirrored in the beta diversity analyses.

Although both comparisons (alpha and beta diversities) show significant differentiation between sites, the ordination of sites is closer for functions than for taxa (Fig. 4). This makes sense, since there are high level metabolic pathways that need to be performed in all microbial mats irrespective of their environment, such as photosynthesis, sulfate reduction
and methanogenesis (*Gutiérrez-Preciado et al., 2018*). However, they still vary enough to show significant differences in terms of the taxa that perform them and the genes involved (*Lloyd et al., 2001*; *Liu & Whitman, 2008*; *Blankenship, 2021*). While taxonomic diversity is strongly influenced by environmental conditions, the functional diversity of microbial mats appears to be more resilient, reflecting the essential metabolic pathways conserved across diverse habitats.

The lack of statistical correlation between pH and temperature with both taxonomic and functional diversity (Table S3) was unexpected given that these variables are known to influence the structure and diversity of microbial communities, including some microbial mats at the local scale (*Peimbert et al., 2012*; *De Anda, Zapata-Peñasco & Souza, 2018*; *Uribe-Lorío et al., 2019*). There are several factors that could have weakened the correlation: 1) Even though the environmental measurements are reportedly taken exactly from the same site as the mats, particular sampling and management methods and logistics may introduce noise; 2) Phototrophic mats can exhibit different pH and temperature dynamics compared to the overlaying water from where the measurements were taken, as well as oscillations throughout the day and seasons (*Jørgensen et al., 1979*; *Bolhuis, Cretoiu & Stal, 2014*); 3) In a recent study on Archaean Domes spatial differentiation (*Espinosa-Asuar et al., 2022*), the most abundant taxa were found to be subjected to environmental filtering associated with salinity levels, while the highly diverse rare biosphere (taxa with a relative abundance <0.01 representing 20% of the sample) did not, suggesting microbial mats can have large reservoirs of low abundance resilient microorganisms that are less responsive to environmental factors.

We found a moderate yet significant correlation between geographical distance and functional structure (*i.e.*, closer mats have a more similar functional composition) (Fig. 7). This result points towards migration and selection of functionally compatible taxa on geographically closer communities. However, there is little to no information about migration of taxa between microbial mats. Future studies focused on migration of taxa between microbial mats will be needed to better interpret this result. Nonetheless, this correlation along the significantly different taxonomic composition found across mats, indirectly echoes the previously studied effects of environmental filtering for phylogenetically conserved functional traits in microbial mats (*Bonilla-Rosso et al., 2012*).

Through the correlations between KEGG pathways, we identified two contrasting clusters of highly abundant functions: biosynthesis and degradation (Fig. S3, Table S2). These clusters are positively correlated within and negatively correlated between each other, pointing to ecological specialization. Ecological specialization of microbial mats along significant differences in diversity and in their taxonomic and functional composition has been previously observed in the red and green oligotrophic mats from Cuatro Ciénegas where, depending on their environment, two contrasting strategies were detected, one mainly based on autotrophic primary production (specialists-dominated mat) and the other on very efficient heterotrophic recycling (generalists-dominated mat) (*Bonilla-Rosso et al., 2012*; *Peimbert et al., 2012*). These strategies coincide with our functional clusters, as biosynthesis includes primary production (photosynthesis and carbon fixation) and degradation includes nutrient recycling. Altogether, the significant

differentiation in the overall metabolism we found in worldwide microbial mats suggests functional specialization as a response to particular environmental conditions.

## Beyond the environment

Physicochemical environmental conditions are not the only factor shaping microbial communities. Biotic interactions and niche dynamics also play a critical role, more so on metabolically interdependent microbial mats (*Paerl, Pinckney & Steppe, 2000*; *Villanueva, Del Campo & Guerrero, 2010*). Here, we found signals that indirectly point to factors beyond the influence of physicochemical environmental variables in the prevalence of dominant taxa, correlation between groups of taxa across sites and functional redundancy.

*Cyanothece*, *Microcoleus* (Cyanobacteria), *Planctomyces* and *Rhodopirellula* (Planctomycetes) were highly prevalent in microbial mats across the globe (Fig. S3) irrespective of temperature and pH. Cyanobacteria is a known dominant taxon in microbial mats which often serves the role of primary producer and structural support (*Bolhuis, Cretoiu & Stal, 2014*), and our results support its role as a keystone species. Moreover, our findings add to the growing body of evidence pointing towards Planctomycetes being an important member of microbial mats (*Davis & Moyer, 2005*; *Steven et al., 2011*; *Fernandez et al., 2016*; *Rozanov et al., 2017*; *Santoyo, 2021*), closely interacting with many other bacterial and archaeal groups (*Saghaï et al., 2017*), although further research is needed to delineate their ecological role.

Similar interspecific interactions were found across all mats, regardless of their environmental differences. We found six clusters of abundant bacteria that are highly correlated across mats from different sites (Fig. 6): cluster 1 mostly includes sulfate-reducing, green sulfur and non-sulfur bacteria and spore-forming generalists; clusters 2, 3 and 4, including metabolically flexible generalist bacteria with large genomes; cluster 5 that is entirely composed of Cyanobacteria; and cluster 6, which encompasses mostly nitrogen-fixing and halophilic bacteria.

Groups and even layers of microbial mats are often highly codependent of each other due to a mix of high specialization, complementary metabolic pathways resulting from gene loss and microbial cross-feeding (*Des Marais, 2003*; *van der Meer et al., 2003*; *Souza & Eguiarte, 2018*; *Sánchez-Pérez et al., 2020*). Of particular interest is that the clusters we found are not mutually exclusive, which means they tend to occur regardless of the environmental influence. Moreover, as already described above, the clusters contain bacteria that perform similar ecological functions, which also aligns with the functional redundancy of microbial mats (*Green et al., 2008*; *Babilonia et al., 2018*; *De Anda, Zapata-Peñasco & Souza, 2018*).

The detected clusters also give away information on the community structure of microbial mats. For instance, strong positive correlations within Cyanobacteria (cluster 5) in the top layer of microbial mats have previously been observed, along with negative correlations with Chloroflexi and anoxygenic phototrophs from deeper layers (cluster 1), as well as strong correlation between Spirochaetes and Deltaproteobacteria (cluster 1), which reflect the stratified nature of the mat (*Saghaï et al., 2017*).
It is noteworthy that clusters 1 and 5 were significantly more abundant in Little Hot Creek while cluster 6 was more abundant in Rottnest Island (Fig. 5), which also highlights the ever present influence of the environment in the community structure of microbial mats. Similarly, other taxa from the clusters were found differentially abundant in particular sites, with their ecological roles aligned with the environmental conditions of the site: taxa related to methane and xenobiotic metabolism in the polluted mines (*Drewniak et al., 2016*), which are similarly abundant in other microbial mats from caves (*Martin-Pozas et al., 2023*); to sulfur metabolism in the sulfur-infused Archaean Domes (*Medina-Chávez et al., 2023*); marine bacteria in the coastal Schiermonnikoog (*Bolhuis, Fillinger & Stal, 2013*); and *Clostridium* in Mono Lake, previously found to be a dominant taxa there (*Humayoun, Bano & Hollibaugh, 2003*) (Fig. 5).

There is a clear interplay between environment and biotic interactions: groups of highly correlated bacteria found in all sites (biotic interactions) appear differentially abundant between sites given the local environmental conditions (environmental filtering). Our findings provide a valuable background and a starting point for more specialized studies to accurately describe the complex ecological interactions happening within microbial mats.

Functional composition shows strong signs of resilience. Functional profiles of KEGG pathways are notably similar across mats (Fig. 2), with nucleic acid and amino acid metabolisms being the most prevalent functional categories. Additionally, alpha diversity of functions among mats is also very similar (Fig. 3), suggesting a general degree of functional diversity that can be expected regardless of the mat habitat. The dominance of amino acid metabolism has been previously observed in hypersaline and arctic microbial mats, and may be attributed to their ecological role as nutrient recycling systems (*Varin et al., 2010*; *Bonilla-Rosso et al., 2012*; *Wong, Ahmed-Cox & Burns, 2016*). These remarkable functional similarities not only highlight important environmental-independent characteristics of microbial mats, but also point to a particular functional profile distinct from other soil and sediment communities (*Zhou et al., 2019*; *Wang et al., 2020*).

## Environment *vs.* interactions: perspectives for future ecological studies in modern microbial mats

Physicochemical environmental conditions allow certain taxa to settle and draw competitive advantages due to founder effect, setting the ground rules for the assembly of the rest of the mat (*Lau, Aitchison & Pointing, 2009*; *Jackson, Hawes & Jungblut, 2021*). Thus, as seen in our study, environmental filtering is crucial for understanding dissimilarities in modern microbial mats (*Lau, Aitchison & Pointing, 2009*; *Brislawn et al., 2019*; *Espinosa-Asuar et al., 2022*; *García-Ulloa et al., 2022*). Subsequently, newly arriving species interact with the founder taxa, either by competing, cooperating or coexisting. Once dominant taxa are established, micro-environmental spatial heterogeneity and cyclical environmental fluctuations may help prevent competitive exclusion of the rare biosphere (*Scheffer et al., 1997*), allowing for niche partitioning. This in turn would allow

for functional redundancy to take place, providing resilience to the mat and strengthening microbial interactions in the community (*De Anda, Zapata-Peñasco & Souza, 2018*).

A consequence of niche partitioning in microbial communities is decreased competition and increased taxonomic diversity (*Pernthaler, 2017*; *Lee et al., 2018*; *Omidi et al., 2021*; *Li et al., 2022*). Our finding of highly correlated clusters of taxa that occupy similar ecological roles across microbial mats is a prime example of the prevalence of niche partitioning in microbial mats. Even though these taxa are functionally equivalent, they are able to not only coexist without excluding each other, but likely co-depend, as evidenced by their ubiquitous co-occurrence across mats.

Even though it is not possible to categorize ecological interactions through co-occurrence of taxa alone (*Blanchet, Cazelles & Gravel, 2020*), repeated co-occurrence can strongly suggest the presence of an underlying positive interaction, as previously seen in network analyses of microbial mats (*Saghaï et al., 2017*). Niche partitioning may not only increase the overall diversity of microbial mats, but also their functional redundancy, resilience and stability.

## CONCLUSIONS

Our study sheds light on the diversity and ecological importance of modern non-lithifying microbial mats, which exhibit remarkable adaptability across various environmental conditions. Our findings imply that environmental filtering can drive specialization within microbial mats, and yet various interspecific interactions are maintained across all worldwide mats regardless of the environment, highlighting the interplay between environment and biotic interactions and offering valuable insights into the underlying mechanisms shaping these resilient communities. Overall, our research highlights the potential of microbial mats as an excellent model system for exploring community ecological processes and concepts.

## ACKNOWLEDGEMENTS

Mariette Viladomat Jasso is a doctoral student from Programa de Doctorado en Ciencias Biomédicas, Universidad Nacional Autónoma de México (UNAM). We give special thanks to Dr. Laura Espinosa-Asuar and Dr. Erika Aguirre-Planter for technical assistance, as well as Rodrigo Zorrilla for his help during fieldwork.

### Funding

This study was funded by the DGAPA-UNAM Proyecto PAPIIT IG200319 and IN204822 and Proyecto R20F0009 CEQUA - ANID. Mariette Viladomat Jasso is a doctoral student from Programa de Doctorado en Ciencias Biomédicas, Universidad Nacional Autónoma de México (UNAM), and has received CONAHCYT fellowship 736510. The funders had no role in study design, data collection and analysis, decision to publish, or preparation of the manuscript.

## Grant Disclosures

The following grant information was disclosed by the authors:
DGAPA-UNAM Proyecto PAPIIT: IG200319 and IN204822.
Proyecto: R20F0009 CEQUA - ANID.
CONAHCYT fellowship: 736510.

## Competing Interests

Luis E. Eguiarte and Valeria Souza are Academic Editors for PeerJ.

## Author Contributions

- Mariette Viladomat Jasso conceived and designed the experiments, performed the experiments, analyzed the data, prepared figures and/or tables, authored or reviewed drafts of the article, and approved the final draft.
- Manuel García-Ulloa conceived and designed the experiments, performed the experiments, analyzed the data, prepared figures and/or tables, authored or reviewed drafts of the article, and approved the final draft.
- Icoquih Zapata-Peñasco conceived and designed the experiments, authored or reviewed drafts of the article, and approved the final draft.
- Luis E. Eguiarte conceived and designed the experiments, authored or reviewed drafts of the article, and approved the final draft.
- Valeria Souza conceived and designed the experiments, authored or reviewed drafts of the article, and approved the final draft.

## Field Study Permissions

The following information was supplied relating to field study approvals (*i.e.*, approving body and any reference numbers):
SEMARNAT.

## Data Availability

The code used for data analysis and visualization as well as the input files are available at GitHub and Zenodo:

- https://Github.com/MarietteViladomat/Worldwide-modern-microbial-mat-METAGENOMICS

- manuelgug, & MarietteViladomat. (2023). MarietteViladomat/Worldwide-modern-microbial-mat-METAGENOMICS: Modern Microbial Mat Metagenomics (metagenomics). Zenodo. https://doi.org/10.5281/zenodo.8305877

The raw metagenomic reads from Archaean Domes samples are available at NCBI: PRJNA612690

The quality filtered reads are available at MG-RAST: mgp90438.

All other metagenome samples used in this study can be found in MG-RAST:
- Shark Bay, mgp82056
- Little Hot Creek, mgp7509 and mgp9770.
- Schiermonnikoog, mgp83560

- Death Valley, mgp20082
- Mono Lake, mgp19115
- Rottnest Island, mgp84373
- Kowary and Zloty Stok, mgp84373

## Supplemental Information

Supplemental information for this article can be found online at http://dx.doi.org/10.7717/peerj.17412#supplemental-information.

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
