# Peer review of "Metagenomic insight into taxonomic composition, environmental filtering and functional redundancy for shaping worldwide modern non-lithifying microbial mats"

_PeerJ, doi:10.7717/peerj.17412_

## Round 0.1 · original submission · Major Revisions

Dear Dr. Jasso and colleagues:

Thanks for submitting your manuscript to PeerJ. I have now received three independent reviews of your work, and as you will see, one reviewer recommended rejection, while another suggested a major revision. I am affording you the option of revising your manuscript according to all three reviews but understand that your resubmission may be sent to at least one new reviewer for a fresh assessment (unless the reviewer recommending rejection is willing to re-review).

In general, the reviewers wish to see a much-improved manuscript as far as presentation and clarity. There are a lot of suggestions that will help here. There also appears to be literature that needs citing. Both reviewers 2 and 3 feel that conclusions are reached that are not supported by the data. The conclusions should be assessed considering all the appropriate studies in the field. Please address concerns about subjectivity. Also, please clearly define the hypothesis/hypotheses being tested in your study.

There are many minor problems pointed out by the reviewers, and you will need to address all of these and expect a thorough review of your revised manuscript by these same reviewers. I agree with the concerns of the reviewers, and thus feel that their suggestions should be adequately addressed before moving forward.

Therefore, I am recommending that you revise your manuscript, accordingly, taking into account all of the issues raised by the reviewers.

I look forward to seeing your revision, and thanks again for submitting your work to PeerJ.

Good luck with your revision,

-joe

·

Basic reporting

The authors perform a whole shotgun metagenomics analysis of microbial mats collected from Cuatro Cienegas Basin, referred to as the Archean Domes, spanning two different seasonal time-points, and compare the findings by performing a comparative metagenomic study using the publicly available 60 microbial mat metagenomic samples for 7 additional locations from across the globe. This large-scale comparative metagenomic analysis formed the basis of the taxonomic (microbial diversity) and functional explorations of the microbial mats worldwide to assess the similarities and potential reasons for differences between the mats' microbial diversities while accounting for the geographical variability. The authors have performed an extensive analysis using the available datasets and tools available to perform the metagenomic analyses. These findings can serve as a future benchmarking reference to other studies exploring other microbial mats and other environmental microbiome analyses, particularly considering the scarcity of available literature comparing several microbial mats across geographical locations.

Experimental design

The authors have performed a robust and extensive analysis of the available shotgun metagenomic samples from different microbial mats with the typical bioinformatics tools and approaches used for microbiome analyses.
The research question has been well-defined (characterizing the microbial mats from a selected location) and contrasting these mat samples against all those that are publicly available.

Validity of the findings

The findings are reasonable and supported by appropriate discussion sections. However, taking into account the length of this manuscript and ease of reliability, it would be beneficial to eliminate some redundancies in the text and/or create labeled subsections under each major section that are easier and more convenient to follow.

Reviewer 2 ·

Basic reporting

In this manuscript they sequenced microbial mats in CCB then compared them to other microbial mats. The literature citations suffers again for honest reporting and citation of the previous work. This includes Laguna de bacalar, Cenote Azul, Great Salt Lake, Pavilion lake, Shark Bay, Highbourne Cay, Guerrero Negro, Lagunas de Ruidera, Lake Richmond, and various Polar mats etc. These should be mentioned explained in the context of microbialites/stromatolites. General void of citation of the known literature.

Fundamentally they call microbial mats 'relictual,' which the disprove by analysis of 62 sites all over the world. How can they be both rare and relics yet be everywhere? This is fundamentally flawed.

They fail to compare carbonate ppt mats vs. non-lifthify mats in relation to the mats they compare. This is the major biogeochemical cycle they perform in the formation of stromatolites through geological time. This is a major misstep not to mention this or the previous work.

Experimental design

Poor and outdated analysis. Mixed analysis between MGRAST and unknown. Unclear if the data was cross-examined to be compared across sample. Also, publishing unpublished data without clear authorship or mentioned to the individuals that made the data.

Validity of the findings

Unknown and unclear. As well not many findings presented in the paper. Figures/tables are low quality and hard to read.

Reviewer 3 ·

Basic reporting

Mainly good - I have a few specific comments for improvement.

Experimental design

Lines 100-103: These are not really hypotheses and also not tested in the data. Functional redundancy was observed but not evaluated in light of perturbations. And “the nutrient scavenging and recycling abilities, and overall functional cohesion” is not tested in or evaluated in the current paper. Please be sure to accurately justify the study and more specifically state what you will test (if you will test specific hypotheses — what is presented are observations about the data).

Validity of the findings

All underlying data have been provided; they are robust, statistically sound, & controlled.

Additional comments

I reviewed a previous version of this paper and was pleased to see the figures updated so that the results could be evaluated. I still have a few comments:

I find the majority of the discussion to be overly speculative and not related to the data presented (or not novel). For example — “the top three functional categories being nucleotide, carbohydrate and amino acid metabolisms” as has been previously observed. “The possible role of biotic interactions” section is more about environmental data and biogeography (and the data presented cannot really be robustly interrogated for interactions anyway). This section talks about filtering and founder effect but really does not interpret the data with these ideas in mind other than to note a correlation between distance and function. The Discussion spends time on environmental filtering when only two variables (temp and pH) are included and then goes on to explain that temp and pH are probably still important or not. I would encourage the authors to focus on the new and novel parts of their study and edit the discussion to accurately and more succinctly highlight their findings in light of related studies.

Lines 100-103: These are not really hypotheses and also not tested in the data. Functional redundancy was observed but not evaluated in light of perturbations. And “the nutrient scavenging and recycling abilities, and overall functional cohesion” is not tested in or evaluated in the current paper. Please be sure to accurately justify the study and more specifically state what you will test (if you will test specific hypotheses — what is presented are observations about the data).

Figure 6 is still impossible to read and the resolution is too poor for zooming in. I suggest showing a lower number abundant taxa in A (could show the full map in the supplemental). I’m not sure what the overall point of B is since you cannot read it but would also have to look up each individual KO to understand something about function from it.

Lines 340-341: What clusters? This seems highly subjective. Also how can one compare the clustering between two very different data types (e.g. taxonomy vs. function). There are far fewer functions, especially at the level you are evaluating here so wouldn’t one assume there would be clusters (particularly for functions that all microbes must have).

Line 349: Do you mean community structure or mat morphology? Unclear here. Also you have no way of controlling or knowing if the mats were samples the same and those contain similar layers (e.g. from methanogens and sulfate reducers to phototrophs) so how does one really compare these disparate data sets?

Line 371-372: I would argue that pH and temperature are two of the more robust measurements. Isn’t it more likely that temp and pH might not be from the same location as the mat? And/or that temp and pH in the mat are almost always different than in the overlying water (and vary throughout a diel cycle if photosynthetic microbes are present).

Line 377: Where is the “the large and highly diverse rare biosphere” defined?

Line 378: Please define “seed bank”

Line 456-457: Based on what evidence?

Line 458-501 re-state results while being overly speculative (or merely stating that these functions are found in other mats). Its not clear what is new here or worth noting from the study given that the authors suggest that looking at additional mat samples from across the globe would be a valuable study.

Line 509: What roles and what conditions specifically are you referring to here?

---

## Round 0.2 · Major Revisions

Dear Dr. Jasso and colleagues:

Thanks for revising your manuscript. The reviewers still have some concerns regarding your manuscript. Please address these ASAP and resubmit your manuscript.

NOTE: please respond to the concerns of reviewer 2 regrading missing references and the methodological shortcomings they provide.

Good luck with your revision,

Best,

-joe

·

Basic reporting

I thank the authors for their attempt at incorporating the feedback provided by the reviewers. I believe the updated manuscript to be much improved from the previous one. Having said that, there are a few aspects that need to be fixed prior to the manuscript being accepted for publication.

For instance, the Introduction section has a few convoluted paragraphs with potential redundancies in paragraphs 3 (page 4) and 4 (page 5). For ease of readability and better positioning the need for this study, it would be better suited if the authors start the third paragraph with what is known in the literature (i.e. studies focused on particular sites or similar environmental conditions), and then point out the limitations in the existing literature (a limited number of reviews and studies analyzing the microbial mats at a global scale)

Experimental design

- The 'Metagenome selection' subsection under the 'Materials and methods' section (page 8) mentions the following:
"To achieve a robust metagenomic comparison, we verified that all analyzed metagenomes were sequenced with the Illumina platform and submitted as raw reads."
There are a few ambiguities in this statement:
1. Which of the Illumina platforms is being referred to here? The authors need to be explicit and specific about the requirement/s here than just saying “the Illumina platform”. The authors previously mentioned the use of the Illumina MiSeq sequencing platform. If that is a requirement (or any other Illumina platforms), it needs to be said with the required specifics here.
2. Given the availability of several sequencing platforms and metagenomic assembly tools that provide comparable genome/metagenome completeness, the authors need to provide their reasoning for constraining the choice of sequencing platform rather than the metagenome sequencing and assembly quality statistics.

- Page 9. The authors mention the use of the "Last Common Ancestor" algorithm from Wood and Salzberg (2014). The name is incorrect: it should be the Lowest Common Ancestor (LCA) or the Most Recent Common Ancestor (MRCA) algorithm.

Validity of the findings

The authors mention the following on page 9:
"Matches of >25 nucleotides and >65% similarity to a taxonomic group or a function with an E-value of ≤10−5 were considered significant, and therefore included in the analysis."
Although the e-value threshold for significance is fairly stringent, the same can’t be said convincingly for the match length (25 nt) or the sequence identity (65%). The authors need to provide some motivation for the choices of these thresholds and/or cite other works where similar thresholds have been used.

Additional comments

There are a few typographic errors and grammatical inconsistencies in the manuscript, many of which are introduced after the edits made to the manuscript as a part of the current revision. I request the authors to thoroughly proof-read the manuscript (not just the new edits) to avoid these errors.

Reviewer 2 ·

Basic reporting

Basically the same as before. They use datasets from MG-Rast then dont cite the papers were the data comes from. MG-Rast has known issues with taxonomic measurement of shotgun data. 25 bp with >65% similarity with an e-value of 10-5 across different sequencing technologies over time? This analysis is again old and not the current state of the art.

Experimental design

poor experiment design once again. Not using contigs assemblies. Using short reads with poor cut offs.

Validity of the findings

Can't tell.

Reviewer 3 ·

Basic reporting

The authors have addressed previous comments by adding references and background / context while updating or removing the speculative statements in the Discussion. I would however strongly encourage the authors to interpret the data and present those findings in the Discussion and, in doing so, avoid "We believe" statements.

Experimental design

Methods described with sufficient detail & information to replicate except how the authors decided that mats were lithifying vs. not in MG-Rast. Its not clear how this distinction was made.

Validity of the findings

All underlying data have been provided.

---

## Round 0.3 · Minor Revisions

The authors have Appealed the earlier decision, and as per PeerJ policies, I was asked to step in and consider the Appeal. Having evaluated it, I believe the Appeal has merit.

I agree that minor revisions are still in order, but I believe that this manuscript is still viable in terms of publication quality and that the authors are trying out a novel approach to their analyses.

1) As Viladomat et al., 2022 is not a peer-reviewed publication, I would drop the reference to it so as not to confuse the reader that it comes from a peer-reviewed publication.

2) I think that in addition to pointing out the microbial mat community differences, more effort should be placed on describing the similarities among these mat communities in terms of both taxa and functionally redundant KEGG metabolism categories (e.g., Figure 2). This is a strong outcome that should be better emphasized in the conclusions.

3) The connections to the evidence presented regarding your hypothesis that the environment is the primary driver in determining mat community structure needs to be better laid out. There are only minimal direct connections presented to this affect (e.g., Figure 7). Please make a stronger case here. In the section entitled, “Beyond the environment”, can you point to any specific taxa-driven ecological interactions affecting community structure? An example or two of these would directly help your argument outlining cause and effect relationship, even if it is circumstantial. Can you point to a specific example in your data of niche partitioning?

4) You need a better analysis of both Figure 5 & 6, not just repeating what is shown in each figure. I do not see anything in the discussion as to why these Figures are important to the analysis or argument.

· Appeal

Appeal


· · Academic Editor

Reject

Dear Dr. Jasso and colleagues:

Thanks for yet again resubmitting your revised manuscript. I have once again received a review from the original second reviewer. This reviewer asserts that much of the original criticism raised in the first round of reviews (as well as subsequent rounds) has still not been addressed. While the other reviewer is more positive, I see the concerns of reviewer 2 as valid.

Specifically, reviewer 2 has seen your manuscript 3 times and finds poor analysis and refusal to cite the literature as major problems. They note your refusal to fix figures, and also note that your work does not demonstrate a grasp of the literature on microbialites/stromatolites/microbial mats. They consider the analysis old and out of date.

Unfortunately, your work will no longer be considered for publication. We must adhere to the scientific review process, and it appears that you are unwilling to modify your manuscript and study according to unbiased peer review.

I believe the issues raised by the reviewers are all valid concerns that need to be seriously considered before moving forward with your work.

Good luck with your study,

-joe

·

Basic reporting

I would like to thank the authors again for their positive reception of the review feedback. I am certain this has improved the overall manuscript in comparison with its earlier submission. I am largely satisfied with the current version of the manuscript. I have a few minor suggestions for improvements and these should be fairly quick to fix.

Experimental design

- The authors need to mention the specific version of the bioinformatics tool used whenever they are referred to for the first time in the manuscript (typically in the Methods section)
For instance, FastQC (line 246), MG-RAST (line 251, the pipeline version is mentioned later but this is the first instance of the tool being referred to in the manuscript), SolexaQC (line 284)

Additionally, the authors need to mention whether any custom parameter settings were used to run any of the referred bioinformatics tools. Generally, it is a good practice to mention something to the effect of the bioinformatics tools used for this study were executed with their default parameter settings unless explicitly specified.

- The statistical testing method "Kruskal-Wallis" is frequently named incorrectly

Validity of the findings

I am satisfied with the modifications made to the manuscript following the review comments from the previous revisions.

Additional comments

There are still quite a few instances of grammatical and typographic errors. It is recommended that the authors proofread their manuscripts multiple times (preferably have it read from someone not on the author list to have it freshly screened and reviewed) to eliminate all avoidable mistakes prior to publication.

Reviewer 2 ·

Basic reporting

They have referenced the datasets correctly and cited them within the manuscript.
This is a basic thing to do that hasn't been done previously.
Many references are still missing context of microbial mats within the introduction.
I would like to see further references to A Decho, JS Foster, P Reid, NR Pace, JR Spear, PT Visscher, LI Falcon, BP Burns, C Saiz-Jimenez, ME Farías, P López-García missing alot of key references here.

Many other microbialite and microbial mat metagenomes have been done previously. I am confused by your selection of just a few. You skip the microbialite/stromatolite work completely.

This paper has been published
Forfeiting the priority effect: turnover defines biofilm community succession
https://www.nature.com/articles/s41396-019-0396-x
Please update the BioRxiv

Lithifying mats can form stromatolites, thromolites, leitolites etc! Not just stromatolites please review that work. Line 60.

Experimental design

The experimental design doesn't make a lot of sense. You choose mats from different environments then stated that they are different - why wouldn't they be? An ancient hot spring mat should be different from an alkaline mat right?

The analysis of MG-RAST again is directly downloaded and out of date. Kraken v1 has known issues for taxonomic. The subsystems are also out of date. Why not download the MG-RAST sequences and do an analysis? eggnog-mapper v2 or metacerberus for KO and functional genes. MetaPhylan2 for taxonomy. They should work well on a even a laptop. Do some pathway enrichments with GAGE/Pathview.

The figures are low quality and not fit for publication.
A github to the MG-RAST pipeline isn't a github link. Put a github page of the code you used.

Validity of the findings

Unclear due to the known issues with MG-RAST
R code not available for review.

Instead of finding the differences which is unclear if there are differences why not find things that are similar between. Shared taxa and pathways.

They are diverse samples but why? Of course, they are different. We can look at them morphologically and see that.

---

## Round 0.4 · Minor Revisions

I strongly advise that you follow the reviewer comments closely this time (including those from the earlier drafts), otherwise I will have to reject your manuscript so that you do not waste reviewer's time. You also need to make sure that your literature citing uses peer reviewed publications.

Reviewer 3 ·

Basic reporting

See below for specific comments.

Experimental design

See below for specific comments.

Validity of the findings

See below for specific comments.

Additional comments

Lines 52-54: There have been many recent studies on the timing and events leading to the evolution of photosynthesis that could be cited here to bolster this statement. I assume this is what is meant by “early phototrophic guilds”.

Lines 89-94: Check this sentence structure — something is amiss here. I think you mean to say temperature and pH are broadly recognized as important drivers of microbial community structure and have been observed to play a role specifically in a few mat studies.

Line 101: Why not “We hypothesize” rather than “We believe” since in 105 you present this as your hypothesis.

Line 317: In the abstract to are careful to point out that you don’t find a relationship with temperature (and again in Line 344). This seems to contradict those statements. Please make sure the message (and interpretations) are all consistent throughout.

Line 344: Another, and more reasonable explanation, is that the temperature and pH are different in the mats. Phototrophic mats will often be different than overlaying water (and will change throughout the day) and the temperature can be different as well (and is likely different between the top and bottom of the mat). Both of these should be mentioned here.

Lines 448-451: These statements seem ambiguous and repetitive about being overly informative. Perhaps make this more specific to your study so that this is useful for reader.

Many of the labels in the figures are too low resolution to read (or overlapping such as figure 4 C and D). Please adjust the labels so that they can be read (or remove them) since I suppose they align with A and B. An option might be to use initials or shortened names to make space for the labels.

---

## Round 0.5 · accepted · Accept

Thanks for making all the thoughtful corrections and additions to the references, this manuscript is much better for it.